# PIC: Revisiting INR for Image Coding with Fast Encoding and Sub-Millisecond Decoding

## Abstract

Implicit neural representation (INR) has achieved remarkable progress in novel view synthesis and image/video coding in recent years. Compared to end-to-end image codecs, INR-based image compressors demonstrate significant advantages in decoding complexity. However, their practical application has been hindered by the inferior encoding speed. Compared to conventional end-to-end codecs, INR-based compressors enjoy markedly lower decoding complexity, yet their adoption in practice remains limited due to slow encoding and underutilized decoding efficiency. In this work, we propose an end-to-end INR image coding architecture, Practical INR Image Codec (PIC), that computes all the necessary information for INR network in a single forward pass, achieving an encoding speed of 20 FPS. Additionally, we implement a highly optimized decoder that reaches 2000 FPS decoding speed, significantly surpassing JPEG's performance at comparable rate-distortion (RD) performance. To the best of our knowledge, this work presents the first learning-based image codec that simultaneously outperforms or is comparable with JPEG in both RD performance and decoding speed while maintaining practical encoding speed. Code will be released.

## 1 Introduction

Image compression has long been a fundamental topic in signal processing and remains a critical technology underpinning more complex systems such as video coding. With the rapid expansion of Internet data, industrial data, AIGC data, and other forms of data, compression technology remains a critically important research area. Image compression technology has undergone significant evolution, encompassing traditional encoders such as JPEG (Wallace, 1992), end-to-end models (Ballé et al., 2017; 2018), and recently emerged compression methods utilizing implicit neural representation (INR) (Dupont et al., 2021; 2022; Ladune et al., 2023) or Gaussian Splatting (GS) (Kerbl et al., 2023; Zhang et al., 2024). Each of these approaches demonstrates distinct strengths and limitations in different aspects of image compression tasks. Traditional image encoders typically leverage human understanding of signals and visual perception, employing predefined rules to discard visually insignificant information for compression. For instance, JPEG exploits the human eye's reduced sensitivity to certain high-frequency components by decomposing the image into different frequencies using Discrete Cosine Transform (DCT) and selectively retaining the most perceptually critical parts, thereby achieving highly efficient compression. Additionally, its codec design strikes a balance between rate-distortion (RD) performance and encoding/decoding speed. Subsequent traditional encoders further improved RD performance by adopting more sophisticated transformation techniques, such as wavelet transforms (Skodras et al., 2001) and predictive coding (Bellard, 2018), although at the cost of increased computational complexity and slower processing speeds.

Classic end-to-end image compression algorithms (Ballé et al., 2017; 2018) conceptually follow the transform coding paradigm used in traditional image codecs. The key difference is that traditional compressors incorporate numerous manually designed transforming rules, whereas the end-to-end approach employs data-driven method to learn the transformation. Taking advantage of the powerful learning capability of neural networks, this data-driven nonlinear transformation can effectively model the image distribution prior to the data, thereby achieving RD performance that surpasses traditional image codecs (He et al., 2022b; Jiang et al., 2023). Furthermore, significant progress has been made based on generative models (Mentzer et al., 2020), particularly in low bit-rate compression scenarios oriented to perceptual metrics (Xia et al., 2025).

Table 1: Comparision of different paradigm. E2E and RM are abbreviation for end-to-end and representation model respectively. Feed forward means the method is able to encode an image in one forward pass, instead of a full training process. RD represents rate distortion performance. We selected three representative methods, Factorized (Ballé et al., 2017), Hyperprior (Ballé et al., 2018), COIN (Dupont et al., 2021), Cool-Chic v4.2 (fast) (Ladune et al., 2023; Leguay et al., 2023; Kim et al., 2024) and GaussianImage (Zhang et al., 2024), for comparison. BD-Rate are calcualted relative to JPEG on Kodak dataset. Other results are representative value from the same experiment. Due to the lack of a fair FLOPs calculation method, the complexity of GaussianImage is left blank. Detailed numerical results of all metrics are shown in Fig. 3 and Fig. 6.

| Model | Paradigm | Feed forward | BD-Rate↓ | Enc.[ms]↓ | Dec.[ms]↓ | FLOPs/pixels↓ |
|---|---|---|---|---|---|---|
| Factorized | E2E | √ | -47.34% | **24.3** | 35.7 | 42.175K |
| Hyperprior | E2E | √ | -57.58% | 141 | 169 | 45.546K |
| COIN | RM | | 25.53% | $1.40 \times 10^6$ | 3.18 | 29.057K |
| Cool-Chic v4.2 | RM | | **-64.86%** | $1.77 \times 10^5$ | 216 | 1.303K |
| GaussianImage | RM | | 36.55% | $6.99 \times 10^5$ | 0.439 | - |
| PIC (Our) | E2E RM | √ | -12.78% | 28.6 | **0.418** | **1.074K** |

More recently, representation-based methods—such as INRs (Dupont et al., 2021) and 3D Gaussian Splatting (Kerbl et al., 2023)—have emerged as promising alternatives. Their key advantage lies in extremely low decoding complexity (Ladune et al., 2023; Liu et al., 2024), often achieving orders-of-magnitude speedups over neural codecs (Zhang et al., 2024). Some variants also rival traditional codecs in rate-distortion performance (Kim et al., 2024). However, these gains come at the cost of extremely slow encoding, often requiring minutes or hours to train a model per image, severely limiting real-world deployment.

In this paper, we propose a new image compression paradigm, Practical INR Image Codec (PIC), which bridges the gap between these paradigms. PIC directly produces a low-bit-rate neural representation in a single forward pass through an end-to-end trained network, combining fast encoding, ultra-fast decoding, and competitive rate-distortion performance. Table 1 demonstrates the main differences among three paradigms.

The primary contributions of this work are summarized below:

- We propose a novel image coding paradigm PIC that directly generates low-bit-rate neural representation models through neural networks, simultaneously achieving fast encoding, ultra-fast decoding, and comparative RD performance.

- We have implemented an optimized decoder that fully translates the low-complexity characteristics of neural representation models into practical high decoding speeds.

- Through comprehensive experiments, we validate the performance of our proposed method, demonstrating significant improvements in both RD performance and encoding/decoding speed compared to prior representation-based image coding approaches. Notably, our method surpasses nvJPEG in terms of decoding speed.

## 2 RELATED WORK

### 2.1 END-TO-END IMAGE COMPRESSION

Classic end-to-end image compression extends transform coding paradigm, which use neural network as both analysis transform and synthesis transform (Ballé et al., 2017). An important feature that distinguishes compression models from other models is the integer symbol constraints in entropy coding, which introduces non-differentiable quantization in training (Guo et al., 2021). Ballé et al. (2017) pioneered a method to jointly optimize reconstruction loss and bit-rate constraints by maximizing the Evidence Lower Bound (ELBO) and formalizes the compression model optimization task as

$$\mathcal{L}_{\phi_g,\theta_g} = \lambda R + D(g_s(Q(g_a(\boldsymbol{x};\phi_g));\theta_g),\boldsymbol{x}), \tag{1}$$

where $g_a(\cdot;\phi_g)$ and $g_s(\cdot;\theta_g)$ are analysis transformation and synthesis transformation respectively. $Q(\cdot)$ is quantization operation. To achieve better task performance, several previous works have also

explored many differentiable approximation of quantization (Ballé et al., 2017; Guo et al., 2021). $R$ is estimated bit rate. $\lambda$ balances the reconstruction quality and bit-rate.

This fundamental architecture has been extensively developed in subsequent research. One important approach is to explore more efficient network architectures. Ballé et al. (2018) introduced scale hyperpriors to better model latent distribution. More work investigate auto-regressive structure (Minnen et al., 2018; Minnen & Singh, 2020; He et al., 2022a) or Transformer model (Zou et al., 2022; Liu et al., 2023). Generative models represent another important category. Mentzer et al. (2020) leveraged GAN architectures for high-fidelity reconstruction. With the rise of diffusion models, more research has begun exploring compression in extremely low bit-rate scenarios (Xia et al., 2025).

Although end-to-end methods are limited in practical applications due to computational constraints, their single forward encoding capability offers distinct advantages over INR training-based encoding, and provides valuable inspiration for exploring similar INR encoding schemes.

## 2.2 Representation-based Image Compression

These representation-based methods can be further categorized into several types, with the most prominent paradigm being the direct mapping of positional coordinates to target spaces. For instance, NeRF (Mildenhall et al., 2020) maps ray angles to density and color along the ray, which has significantly impacted the fields of volume rendering and novel view synthesis, bringing widespread attention to INR technology (Barron et al., 2022). Subsequently, this paradigm has expanded to other signal representation domains, including neural rendering (Sztrajman et al., 2021), image representation, video representation, and further into image and video compression (Dupont et al., 2021; Chen et al., 2021).

Another approach involves using learnable parameters or grid as inputs instead of directly utilizing coordinates. Within the NeRF series, Instant-NGP (Müller et al., 2022) is a representative work that employs a multi-resolution spatial hash grid to store these learnable parameters. This same concept can be extended to other neural representation tasks. There are many workss have made remarkable progress in representing and compressing textures (Vaidyanathan et al., 2023), BRDF (Dou et al., 2024), etc. Similarly, the COOL-CHIC (Ladune et al., 2023) and its successors (Kim et al., 2024) have demonstrated impressive performance in image compression. This methodology also finds broad applications in video compression (Kwan et al., 2023).

The 3D Gaussian Splatting (3DGS) (Kerbl et al., 2023) provides a different perspective of representation models. By directly organizing learnable parameters through specific structures, without relying on per-instance overfitted neural networks, 3DGS has demonstrated unique advantages in novel view synthesis tasks. Compared with INR, this representation method often exhibit more interpretable structure, providing unique advantages beyond quantity performance. Similarly, such methods have been successfully applied to tasks including 3D scene representation (Lu et al., 2024; Huang et al., 2024), image compression (Zhang et al., 2024), and video compression (Liu et al., 2025).

This class of methods has demonstrated unique advantages in compression, such as low decoding complexity, fast decoding speed, and high reconstruction quality. However, these strengths are difficult to achieve simultaneously in a single model. Moreover, the reliance on model training for encoding hinders their practical deployment. In this work, our proposed method explores how to balance multiple metrics to construct a practical encoder.

## 3 Method

The primary reason for the slow encoding of representation models lies in the fact that the encoding process itself is the training process. Consequently, methods based on implicit neural representations—as well as similar approaches like Gaussian splatting—can only be applied in scenarios where encoding time is highly insensitive. A straightforward idea is to directly generate this representation model itself through a neural network. The proposed PIC follows the idea. Fig. 1 demonstrates the overall architecture of our method. Section 3.1 shows the details of transforming a representation to

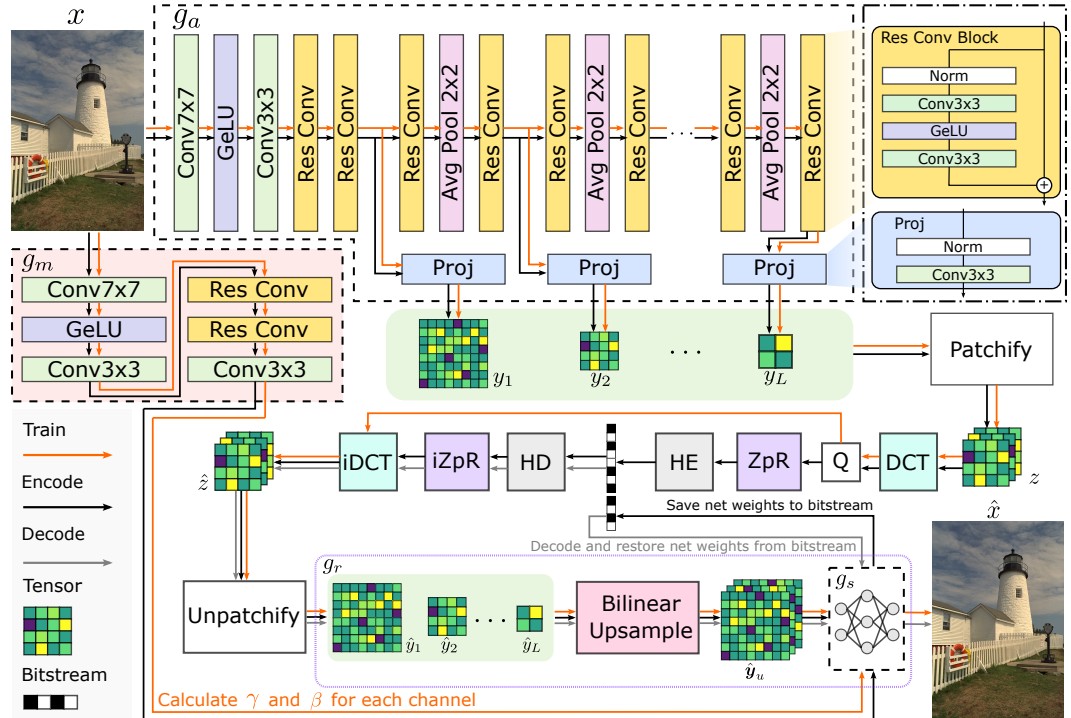

Figure 1: The framework of PIC. The overall data flow is presented in an S-shaped in the diagram, following $g_a \to \hat{\boldsymbol{y}} \to z \to \hat{z} \to g_r \to g_s$. $g_a$ and $g_m$ are latent encoder and modulation net respectively. $y_1, y_2, \ldots y_L$ are latents generated by latent encoder $g_a$. All latents are divided into patches of size $8 \times 8$ and then concatenated together as $z$. Similar to other compression methods, $z$ is quantized after DCT transformation. The ZpR module converts the quantized symbols into a more compact form, and we will discuss this module in detail in the Section 3.3. HE and HD are Huffman encoder and decoder. iZpR and iDCT are inverse transformation of DCT and ZpR respectively. $g_r$ is part of the decoder and also acts as an image representation model. $g_s$ is synthesis network. $\gamma$ and $\beta$ are mean and standard deviation of each channel of $g_m$ output respectively. Note unlike ARM model in Cool-chic-style decoding, we directly employs Huffman decoding to obtain all latents in decoder.

compression models. Section 3.2 introduces entropy estimation module. Section 3.3 describes the full pipeline and implementation details.

## 3.1 Repurposing Representation Models for Compression

To better introduce the proposed method, we begin with the image representation model $g_r$ in Fig. 1. For a $H \times W$ image, $\hat{\boldsymbol{y}}$ is a set of pyramid-like multi-resolution latents

$$\hat{\boldsymbol{y}} = \{\hat{y}_i \in \mathbb{R}^{H_i \times W_i}, i = 1, 2, \ldots, L\}, \tag{2}$$

where $H_i = \frac{H}{2^{L-i}}, W_i = \frac{W}{2^{L-i}}$. $L$ is the number of latents. $g_s$ is a simple MLP. To match the resolution of the output image, $\hat{\boldsymbol{y}}$ is upsamlped to $\hat{\boldsymbol{y}}_u \in \mathbb{R}^{C \times H \times W}$ before being fed into the reconstruction network $g_s$. If we apply $g_r$ in image representation task, image information will be stored in the weights of the network. All parameters, including $\hat{\boldsymbol{y}}$ and weights in $g_s$, are trainable and optimized jointly via gradient descent.

Obviously, achieving a fast encoding process through training is highly challenging. Therefore, our approach generates all weights of the representation model in a single step. Since the latents inherently lies in a space similar to the image domain, designing a simple yet functional encoder $g_a$ is relatively straightforward, as shown in Fig. 1. The primary challenge is generating the network weights of $g_s$. While previous works have explored methods for network weight generation, their

performance in compression tasks still leaves room for improvement (Chen et al., 2024). In this paper, we adopt a simple strategy called channel-wise normalization and modulation.

We first divide $g_s$ into shared part $\mathrm{MLP}^s$ and instance-dependent part $\mathrm{MLP}^i$. Suppose $N$ is batch size and $\boldsymbol{f} \in \mathbb{R}^{N \times C \times H \times W}$ is the intermediate feature between $\mathrm{MLP}^s$ and $\mathrm{MLP}^i$, the instance-normalized feature is

$$\bar{\boldsymbol{f}} = \frac{\boldsymbol{f} - \mathrm{mean}(\boldsymbol{f})}{\mathrm{std}(\boldsymbol{f})}. \tag{3}$$

Then modulate the normalized feature $\bar{\boldsymbol{f}}$

$$\tilde{\boldsymbol{f}} = \gamma \odot \bar{\boldsymbol{f}} + \beta, \tag{4}$$

where $\gamma \in \mathbb{R}^{N \times C}$ and $\beta \in \mathbb{R}^{N \times C}$ are generated by modulation net $g_m$. Fraction and $\odot$ represent element-wise division and multiplication at $C$ dimension respectively. Note $g_m$ will generate corresponding $\gamma$ and $\beta$ for each input $x$. One notable advantage of this transformation is that both the normalization and modulation parameters can ultimately be fused into the network weights of $\mathrm{MLP}^i$, thereby achieving the goal of generating corresponding network weights for each input. For detailed derivation, please refer to the appendix A.1.

## 3.2 ENTROPY ESTIMATION

In neural representation models, constrained by the overall model size, it is challenging to incorporate a large entropy estimation network. One solution is to employ a compact neural network to enhance RD performance via auto-regressive methods (Ladune et al., 2023). However, the decoding speed of such approaches remains limited by their auto-regressive design. Inspired by Luo et al. (2020), we estimate the distribution of DCT parameters to achieve rate estimation accordingly.

For latents generated by $g_a$

$$\boldsymbol{y} = \{y_i \in \mathbb{R}^{H_i \times W_i}, i = 1, 2, \ldots, L\}, \tag{5}$$

we divide them into patches of size $8 \times 8$, concatenate together as $z \in \mathbb{R}^{n \times 8 \times 8}$ and transform to symbols through DCT

$$z = \mathrm{Patchify}(y_1, \ldots, y_L), \tag{6}$$

$$s = \mathrm{DCT}(z). \tag{7}$$

Let $s_{i,k}$ is the $k$-th DCT component ($k \in \{0, \ldots, 63\}$) of $i$-th block, the estimated entropy is

$$\mathcal{L}_{\mathrm{entropy}} = -\sum_{i=1}^{n} \sum_{k=0}^{63} \log_2 p_{\theta_k}(s_{i,k} + u), u \sim \mathrm{Uniform}(-0.5, 0.5). \tag{8}$$

We use 64 channels entropy model, which means we model the distribution of DCT components independently. $\theta_k$ is the corresponding parameters for the $k$-th channel in entropy model. Similar to other compression framework , the entire pipeline is non-differentiable after quantization of $s$ , so we use a simple uniform noise relaxation (Ballé et al., 2017) to build an end-to-end trainable pipeline.

## 3.3 FULL PIPELINE AND HARDWARE-AFFINITY IMPLEMENTATION

JPEG reduces bit-rate through lossless compression techniques like the zigzag transform and run-length encoding (RLE). In our PIC, we similarly employ **Z**igzag reordering and **p**artial **R**un-length encoding (ZpR) module to improve RD performance. As shown in Fig. 2, ZpR reduce zeros in zigzag reordered symbols. The main difference between ZpR and RLE in JPEG is we only reduce zeros in symbols. This strategy is more straightforward to implement while effectively reducing the bit-rate. We also optimize the overhead of code table, as described in Appendix A.2.1.

Fig. 1 illustrates the full pipeline of PIC. We integrated all of the above modules to achieve an trainable codec. The final loss function is

$$\mathcal{L} = \lambda \mathcal{L}_{\mathrm{entropy}} + D(x, \hat{x}), \tag{9}$$

where $D$ is distortion metric, $\lambda$ balances the trade-off between reconstruction quality and bit-rate.

Figure 2: The pipeline of ZpR module. ZpR transform quantized DCT coefficients to compact symbols through three steps: zigzag reorder, trim trailing zeros and encode zeros using run length encode (partial RLE).

Another important topic is how to implement an optimized decoder. JPEG's concise design and long-term optimizations have made it one of the fastest encoders. However, due to its early introduction, JPEG has not fully leveraged the hardware advancements in recent years. Benefiting from recent research on parallel Huffman decoding (Weißenberger & Schmidt, 2018) and the emergence of hardware features like Tensor Cores, it has become possible to develop a decoder that surpasses JPEG in performance.

In architecture design, we avoided using relatively large synthesis network like COOL-CHIC-family (Ladune et al., 2023; Kim et al., 2024) to comply with Tensor Core's matrix structure requirements. Each layer in our reconstruction network $g_s$ maintains a width of 16, precisely matching the warp matrix multiplication of TF32 operations in Tensor Cores. This alignment maximizes hardware utilization and accelerates reconstruction speed. It should be noted that, although our current implementation relies on NVIDIA's Tensor Cores, other hardware also supports similar types of functionalities, such as AMD's rocWMMA. We believe such acceleration hardware will eventually be supported by more manufacturers' devices in the future as well.

Furthermore, since the precision of TF32 is lower than standard FP32, a gap would arise if training were conducted with FP32 while inference/decoding relied on Tensor Core-accelerated operations. To address this, we draw inspiration from neural rendering techniques and implement the training-phase code using the differentiable shading language Slang (He et al., 2018), which significantly reduces the complexity of integrating inline CUDA code in differentiable framework. This approach ensures consistency between training and inference while maintaining computational efficiency during training.

Qualitatively speaking, our decoder's complexity is actually higher than JPEG's. However, by fully leveraging hardware capabilities, our decoder has surpassed commercial decoders and provided new insights for the design of image codecs.

## 4 EXPERIMENTS

### 4.1 EXPERIMENT SETUP

**Dataset.** Since the proposed PIC is a generalizable method, it is necessary to train our model on a large dataset like end-to-end methods. We use LSDIR dataset (Li et al., 2023) as training set, which includes 84,991 natural images. During training, these images were randomly cropped to a resolution of $512 \times 512$. Additionally, Our evaluation is conducted on two popular datasets, Kodak[1] and CLIC[2]. Kodak dataset includes 24 images of size $768 \times 512$. The CLIC dataset contains 41 high-resolution natural images.

**Metrics.** We use the peak signal-to-noise ratio (PSNR) in RGB 4:4:4 as distortion metric, which are the most widely used metric in compression. We also report and MS-SSIM (Wang et al., 2003) LPIPS (Zhang et al., 2018) in main experiment, which is a popular perceptual metric to measure realism. Bit-per-pixel (BPP) is used as coding efficiency metric. To demonstrates the efficiency of PIC, we report encoding and decoding time as well.

---

[1]http://r0k.us/graphics/kodak

[2]https://clic.compression.cc/2021/tasks/index.html

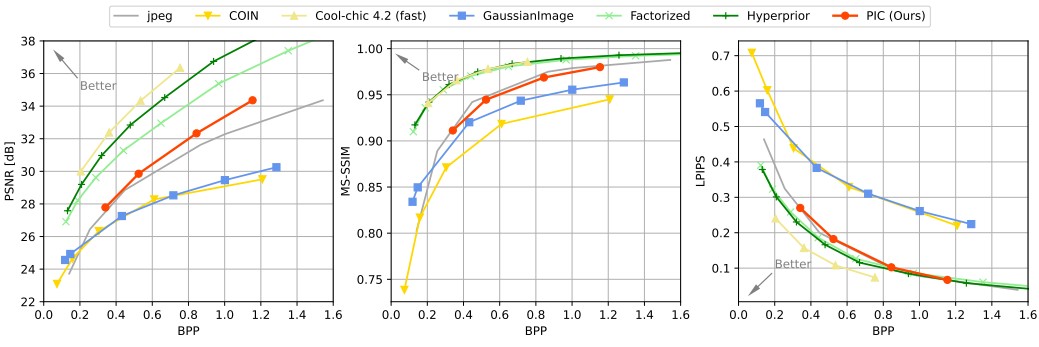

(a) Rate-distortion performance on Kodak dataset.

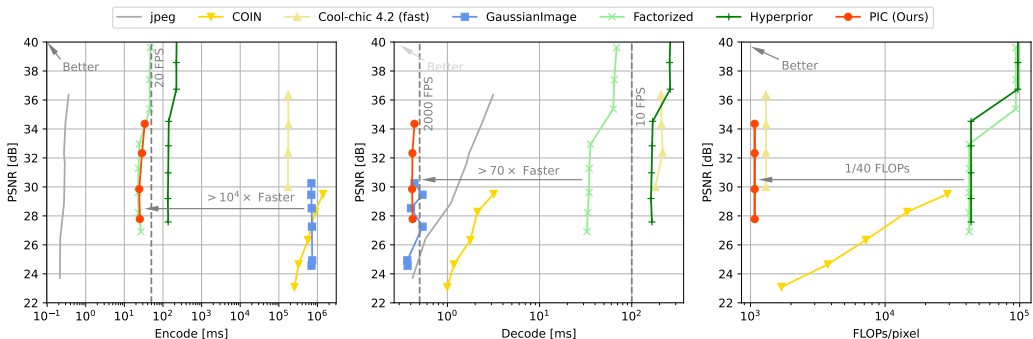

(b) Speed and complexity comparison of all methods on Kodak dataset.

Figure 3: Comparison of all methods. PIC achieves better PSNR at high bit-rate region and comparable MS-SSIM and LPIPS performance with JPEG. PIC also outperforms other representation-based methods with similar decoding speed in all quality metrics at high bit rate region. For speed and complexity comparison, PIC achieves comparable encoding speed with Factorized (Ballé et al., 2017), significantly faster decoding speed and low decoding complexity. Compared to other representation models, PIC has an orders-of-magnitude advantage in encoding speed.

**Implementation Details.** Our complete framework is implemented in PyTorch, with the key distinction that the synthesis network is built upon Slang-torch [3] to simplify Tensor Core programming within the PyTorch environment. The ZpR module and decoder are CUDA-accelerated, while the Huffman codec is adapted and modified from GPUHD (Weißenberger & Schmidt, 2018). Other auxiliary components such as bitstream I/O operations are implemented in C++. All experiments are performed on a single NVidia RTX 4090D. We jointly optimize the full set of parameters in $g_a$, $g_m$ and $g_s$. This optimization is performed separately for each $\lambda = \{0.01, 0.005, 0.002, 0.001\}$ using Adam optimizer and learning rate of 1e-4. All convolution layers have 16 channels. $g_s$ is a 4 layers MLP with ReLU activation. The final bitstream includes header information, Huffman code table, encoded latents, and parameters of MLP$^i$ in FP32 format. More details can be found at Appendix A.2.1 and Appendix A.2.

**Benchmarks.** Our method is benchmarked against competitive representation-based methods like COIN (Dupont et al., 2021) and GaussianImage (Zhang et al., 2024). We also compares with Cool-Chic v4.2 (fast) (Ladune et al., 2023; Leguay et al., 2023; Kim et al., 2024) which is the state-of-the-art method in representation-based image codec. We reproduce all the results using the original source code. Another baselines include pretrained Factorized model (Ballé et al., 2017) and Hyperprior (Ballé et al., 2018) in CompressAI (Bégaint et al., 2020) and nvJPEG[4], which is a CUDA-accelerated JPEG codec. Note we use MSE as distortion metrics in loss function for PIC in all experiments. For other methods, we follow their original loss function settings.

---

[3]https://github.com/shader-slang/slang-torch

[4]https://developer.nvidia.com/nvjpeg

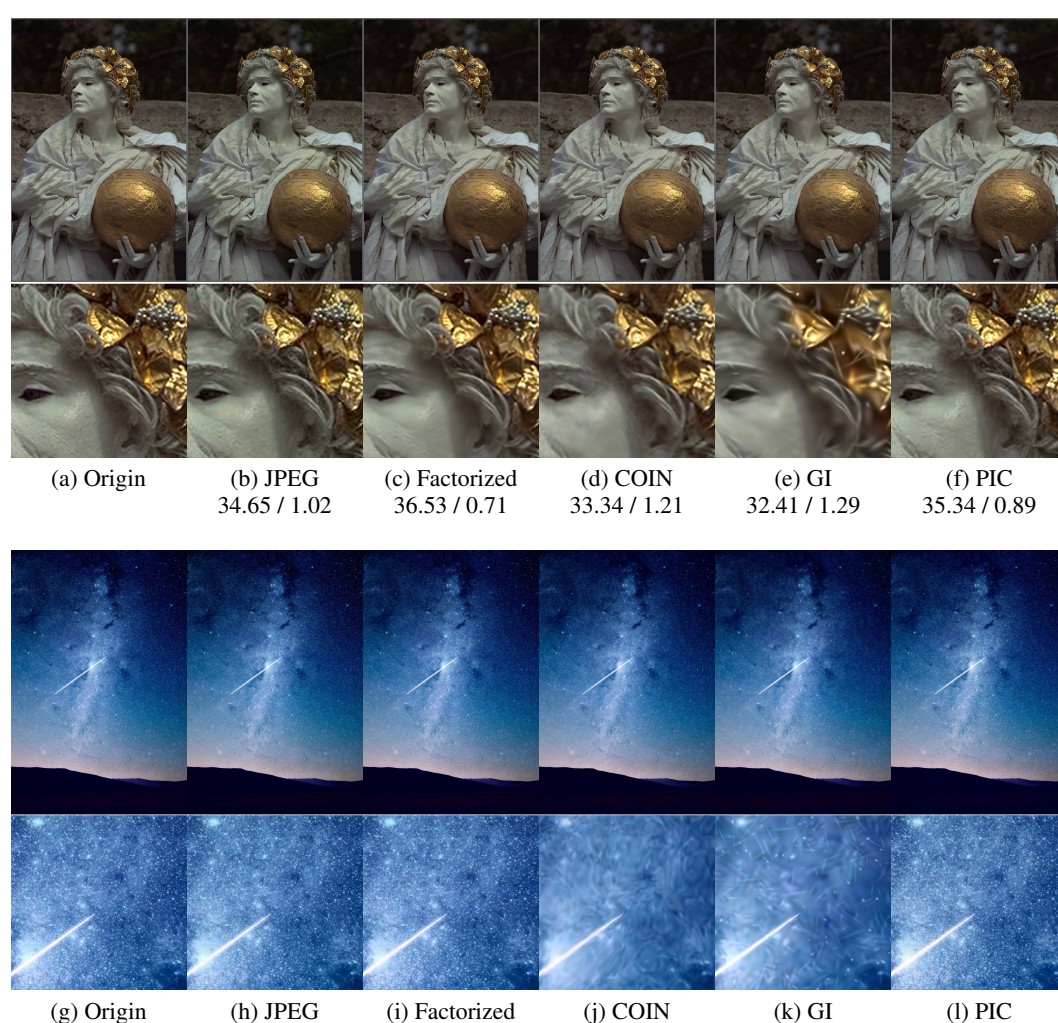

Figure 4: Visualizations on kodim17.png from Kodak dataset and juskteez-vu-1041.png from CLIC dataset of JPEG (Wallace, 1992), Factorized model (Ballé et al., 2017), COIN (Dupont et al., 2021), GaussianImage (abbreviated as GI) (Zhang et al., 2024) and proposed PIC. Below each image are the corresponding PSNR/bpp results. On juskteez-vu-1041.png, we select the model setting with best quality in original paper for COIN and GaussianImage.

## 4.2 QUANTITY AND QUALITY RESULTS

Fig. 3 shows the main results of different methods on Kodak dataset. The experiment results of CLIC dataset can be found in Fig. 6 at Appendix A.3.1. Fig. 3a demonstrates the RD performance of all methods. All results are averaged on same corresponding hyper-parameters setting for each method. PIC significantly outperforms COIN and GaussianImage on all metrics. This advantage becomes more obvious as the bit-rate increases. This not only demonstrates the superiority of our proposed method, but also highlights the unique advantages of grid-based approaches in reconstruction quality. Our experiments further validate the performance advantages of COIN and GaussianImage in the low bit-rate range, while these methods exhibit weaker quality improvement compared to other approaches as the bit-rate increases. However, when compared to Cool-chic, Factorized model, and Hyperprior model, PIC still has significant room for improvement in RD performance. Fig. 3b presents a comprehensive comparison of encoding/decoding speeds across all methods. Our method demonstrates superior encoding speed compared with other representation-based approaches, achieving an acceleration of exceeding three orders of magnitude. In terms of

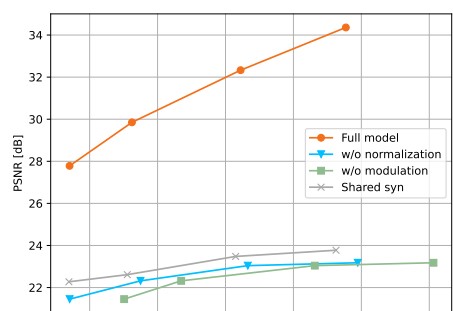 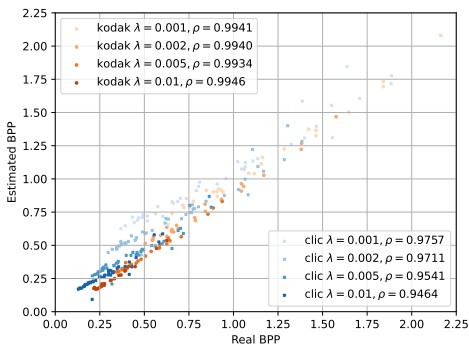

(a) Ablation of synthesis net $g_s$ architecture. All components are necessary for performance.

(b) Ablation of entropy model. $\rho$ is Pearson correlation coefficient.

Figure 5: Ablation results.

decoding speed, our approach ranks second only to GaussianImage while outperforming all other alternatives.

Fig. 4 is the visualization of all methods. Our method demonstrates marginally superior performance compared to JPEG while significantly outperforming COIN and GaussianImage. Our approach also achieves comparable results to JPEG in terms of texture details and artifacts, which can be largely attributed to similar entropy modeling. In contrast, the Factorized model tends to produce overly smoothed outputs. COIN exhibits swirling artifacts, and the GaussiaImage displays Gaussian-like speckled artifacts.

Overall, although there is still a certain gap between our method and state-of-the-art approaches in terms of RD performance, the reality is that current architectures, whether based on end-to-end autoencoder or the overfitted Cool-Chic architecture, are constrained by their inherent limitations, making them difficult to deploy in general scenarios. Even the most lightweight Factorized model is hampered by its high computational overhead. For Cool-chic-like models, their excessively slow encoding and decoding speeds render them impractical for most application. Meanwhile, the proposed PIC exhibits no significant weaknesses across key practical metrics. This offers a new perspective for implementing learning-based codec in real-world scenarios.

### 4.3 ABLATION STUDY

Fig. 5a presents the architectural ablation results of the synthesis network $g_s$. "Shared" denotes using the same synthesis net $g_s$ for all images, in which case our method degenerates into an end-to-end model. Evidently, the model performance under this configuration proves significantly inferior due to the limited capacity of decoder network. W/o normalization and w/o modulation respectively indicate the exclusion of normalization and modulation in $g_s$. Unsurprisingly, such configurations significantly degrade model performance. Only when incorporating all modules does our method achieve the desired performance.

Due to the inherent approximation nature of entropy networks, discrepancies with actual performance are unavoidable. Furthermore, post-processing in the ZpR module may potentially amplify these deviations. Fig. 5b illustrates the differences between the actual bit-rate and the estimated bit-rate. Although some discrepancies exist, overall consistency has been maintained. This finding aligns with prior studies (Luo et al., 2020). Developing more accurate entropy estimation models remains an important direction for future improvements.

We further analyze the latent representation structure to better understand the model's performance. Appendix A.3.2 presents the visulization of latents. The detailed design of the pRLE module and the structural design of the synthesis network are also investigated. Additional results can be found in the Appendix A.3.3 and Appendix A.3.4.

## 5 CONCLUSION

In this work, we introduced PIC, an end-to-end INR image coding framework that generates all network parameters in a single forward pass, leading to an encoder substantially faster than prior representation-based approaches. We further designed a highly optimized decoder that outperforms JPEG in speed while delivering comparable rate-distortion performance. Together, these advances establish a new balance between efficiency and quality, setting a milestone for INR-based compression. While there is still headroom for improving RD performance, our method opens a novel paradigm for practical learning-based image codecs and provides a foundation for extending INR-based compression to other modalities.

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

# A  APPENDIX

## A.1  FUSING MODULATIONS TO SYNTHESIS NET

In the encoding process, both the normalization and modulation parameters are fused into the parameters of $\text{MLP}^i$. The modulated network parameters subsequently included as part of the bitstream. Suppose $\boldsymbol{f} \in \mathbb{R}^{C \times H \times W}$ is the output feature of $\text{MLP}^s$

$$\boldsymbol{f} = \text{MLP}^s(\hat{\boldsymbol{y}}_u), \tag{10}$$

where $\hat{\boldsymbol{y}}_u$ is upsampled latents shown in Fig. 1. Note we omit batch dimensions for clarity. The modulated features is

$$\bar{\boldsymbol{f}} = \frac{\boldsymbol{f} - \mu}{\sigma}, \tag{11}$$

$$\tilde{\boldsymbol{f}} = \gamma \odot \bar{\boldsymbol{f}} + \beta, \tag{12}$$

where $\mu = \text{mean}(\boldsymbol{f}), \sigma = \text{std}(\boldsymbol{f})$. Here we normalize $\boldsymbol{f}$ at $H$ and $W$ channel. Fraction represents element-wise division at $C$ dimension. $\odot$ is element-wise multiplication at $C$ dimension. $\gamma \in \mathbb{R}^C$

and $\beta \in \mathbb{R}^C$ are modulation vectors. The first layer of MLP$^i$ can be expanded as

$$
\begin{align}
\boldsymbol{f}' &= w^T \tilde{\boldsymbol{f}} + b \tag{13}\\
&= w^T(\gamma \odot \bar{\boldsymbol{f}}) + w^T\beta + b \tag{14}\\
&= \frac{w^T(\gamma \odot \boldsymbol{f})}{\sigma} - \frac{w^T(\gamma \odot \mu)}{\sigma} + w^T\beta + b. \tag{15}
\end{align}
$$

The net weight and bias included in bitstream are

$$
w' = \frac{\gamma^T \odot w}{\sigma^T}, \tag{16}
$$

$$
b' = -\frac{w^T(\gamma \odot \mu)}{\sigma} + w^T\beta + b. \tag{17}
$$

In decoding phase, only $w'$ and $b'$ are required, with no need for normalization and modulation. In our experiments, we use first 3 layer as MLP$^s$ and last layer as MLP$^i$.

### A.2 MORE IMPLEMENTATION DETATILS

#### A.2.1 ENCODING HUFFMAN CODE TABLE

Unlike JPEG, our method employs image-specific Huffman coding tables, meaning the storage overhead of the Huffman tables also impacts the final bit-rate performance. To ensure $O(1)$ complexity for decoding one symbol, GPUHD builds a coding table including $2^p$ entries ($p$ is precision) which covers all possible bitstream pattern for decoding one symbol. For small pictures like those in Kodak dataset, the overhead of storing the code tables is non-negligible. Fortunately, the actual number of symbols used in encoding is typically fewer than 128, allowing for further compression of the code tables to reduce this overhead. Benefiting from the prefix code property of Huffman coding, we can apply run-length encoding (RLE) to compress the original code table, reducing the number of table entries to match the symbol count and significantly decreasing the storage overhead of the original code table.

#### A.2.2 FAST CHUNKING ALGORITHMS

Algo. 1 is a fast chunking algorithm that splits the symbol stream based on end-of-block markers (eob_marker) and expands the zero symbols according to the pRLE code table. This function requires precomputing the offsets of all eob_marker to enable parallel processing. We use thrust::copy_if to efficiently obtain all offsets. For synthesis network $g_s$, we fuse all layers in a single CUDA kernel to achieve efficient processing. All matrix multiplications in $g_s$ are implemented using APIs provided by nvcuda::wmma.

#### A.2.3 HYPER-PARAMETERS SETTINGS

For proposed PIC, we use total $L = 8$ latents. This means that the minimum size of images we can encode is $256 \times 256$. This also suggests that encoding smaller images requires reducing the value of $L$. During the training, we set batch size to 16 and randomly cropped the input image to a resolution of $512 \times 512$. We trained our model for 10 epochs in all experiments. Mean squard error (MSE) is used as distortion metric in training.

#### A.2.4 DETAILS OF BASELINE METHODS

For COIN and GaussianImage, We reproduce all the results using the original source code. It should be noted that although our method only used MSE as the distortion metric during the training, we still adhered to the original loss function settings for the baseline methods, even if the original approaches employed loss functions other than simple MSE. Additionally, for all methods, we did not use model versions specifically trained for different test metrics. For instance, in the case of Factorized model, all experiments utilized a pre-trained version optimized with MSE as the loss function.

---

**Algorithm 1** Expand Symbols

---

1: **procedure** EXPAND_SYMBOLS(symbols, offsets, output)
2:     offset_start ← 0
3:     **if** gid ≠ 0 **then**                                             ▷ gid is CUDA thread id
4:         offset_start ← offsets[gid−1] + 1
5:     **end if**
6:     offset_end ← offsets[gid]
7:     eob_marker ← symbols[offset_end]                          ▷ Get eob_marker from symbol stream
8:     output_symbols ← output + gid × 64
9:     offset ← 0
10:    **for** $i$ ← offset_start **to** offset_end−1 **do**
11:        $s$ ← symbols[$i$]
12:        **if** $s \geq$ eob_marker $- 17 + 1$ **then**              ▷ If $s$ is a symbol in the pRLE code table
13:            **for** $j$ ← 0 **to** $s + 17-$ eob_marker $-1$ **do**
14:                output_symbols[ZIGZAG_ORDER[offset]] ← 0
15:                offset ← offset + 1
16:            **end for**
17:        **else**
18:            output_symbols[ZIGZAG_ORDER[offset]] ← symbols[$i$]
19:            offset ← offset + 1
20:        **end if**
21:    **end for**
22: **end procedure**

---

### A.2.5 EVALUATION PROTOCOL

Due to the fact that speed evaluation is susceptible to interference from various factors, including hardware occupancy and PyTorch's inherent asynchronous design, we have carefully designed the speed evaluation process and ensured exclusive access to the hardware during assessment. Although there are currently many implementations of JPEG, in order to achieve a fair comparision, it is necessary to use a GPU-optimized version. However, since nvNVJPE is not open-source, we implemented our binding based on previous implemntation [5]. We have also meticulously optimized the input and output components of nvJPEG to minimize any additional overhead as much as possible. Besieds, we run encoding and decoding in same process without saving bitstream to disk for nvJPEG, which is intended to eliminate the I/O time. In experiment, we find running encoding and decoding in same process is sufficient to warmup nvJPEG. For all method, we report the result of serial decoding all images in corresponding dataset, following is part of benchmark code for nvJPEG

---

**Listing 1** Snippet used to mesure nvJPEG decoding speed.

---

```python
for idx in range(len(bs_pack)):
    jpeg_bytes, img, f = bs_pack[idx]  # encoded bitstream
    torch.cuda.synchronize()
    start_time = time.time()
    img_decoded = coder.decode(jpeg_bytes) # run nvJPEG decoder
    torch.cuda.synchronize()
    end_time = time.time()
    total += end_time - start_time
print('decoding time', total / len(bs_pack))
```

---

We will release the benchmark code.

---

[5]https://github.com/UsingNet/nvjpeg-python

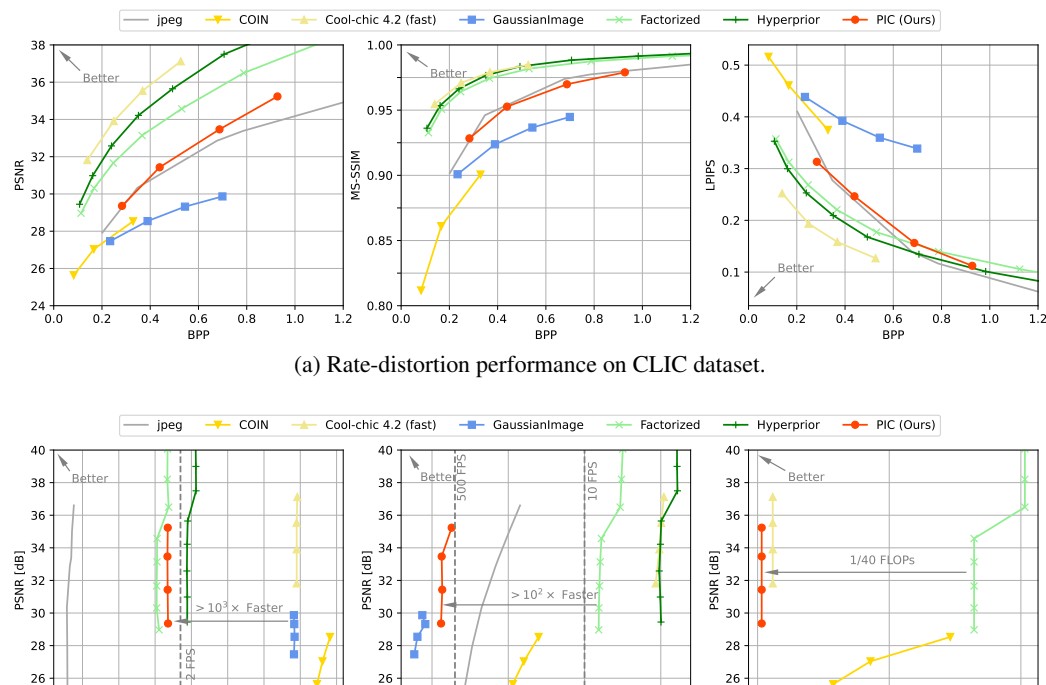

(a) Rate-distortion performance on CLIC dataset.

(b) Speed and complexity comparison of all methods on CLIC dataset.

Figure 6: Results on CLIC dataset.

### A.3 MORE EXPERIMENT RESULTS

#### A.3.1 RESULTS ON CLIC DATASET

Fig. 6 demonstrates full results on CLIC dataset. The overall results are consistent with those on the Kodak dataset, indicating that PIC outperforms previous representation models in comprehensive performance, while also holding advantages over end-to-end models in certain metrics.

#### A.3.2 LATENTS VISUALIZATIONS

To better demonstrate the model's performance under different bitrates, we visualized the latent representations when $\lambda = \{0.01, 0.001\}$. Fig. 7 shows the visualizations without normalization. Overall, the latents at different resolutions reflect varying levels of detail from the original image. Fig. 8 presents the results after applying same normalization to all latents. Although these latents do not directly indicate the magnitude of the corresponding symbols, they indirectly reveal that at smaller $\lambda$, higher-resolution latents retain more details, while at bigger $\lambda$, more details are contained in relatively lower-resolution latents. This also reflects the bits allocation tendencies under different settings.

#### A.3.3 ABLATION STUDY OF MODEL ARCHITECTURE

Given implementation complexity considerations, we opted for a simplified RLE. Because our symbol range is determined by training, we cannot fix an RLE encoding table like JPEG, but dynamically determine it through the transformed symbol stream. Table 2 presents a performance comparison of performing RLE encoding on more symbols (-1 and 1). From the results in the table, it can be seen that performing RLE only on 0 is not only easier to implement in engineering, but can also achieve better results in certain situations.

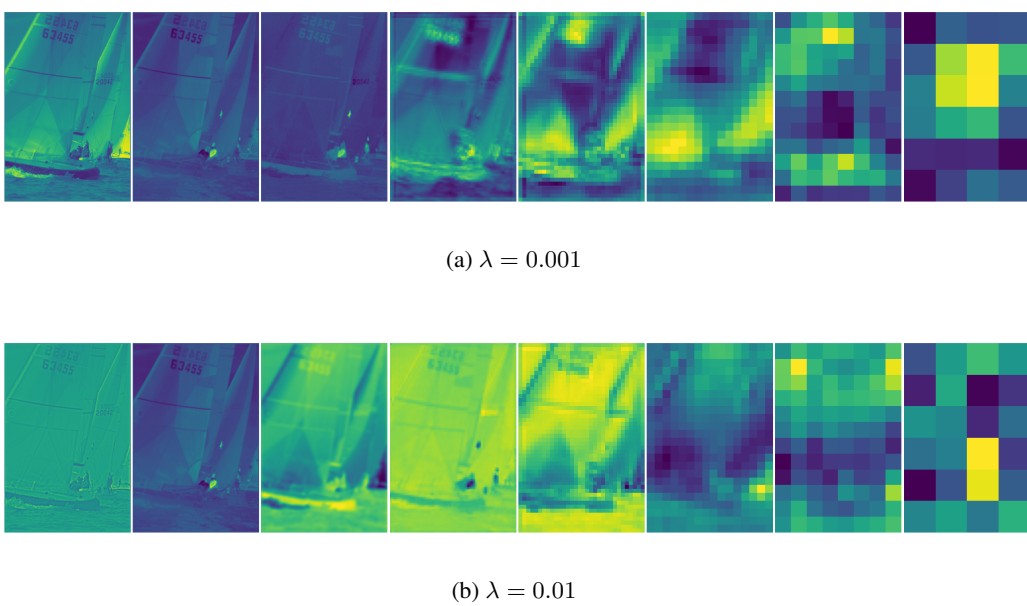

(a) $\lambda = 0.001$

(b) $\lambda = 0.01$

Figure 7: Visualizations of latents. These figures demonstrate unnormalized value for each latent.

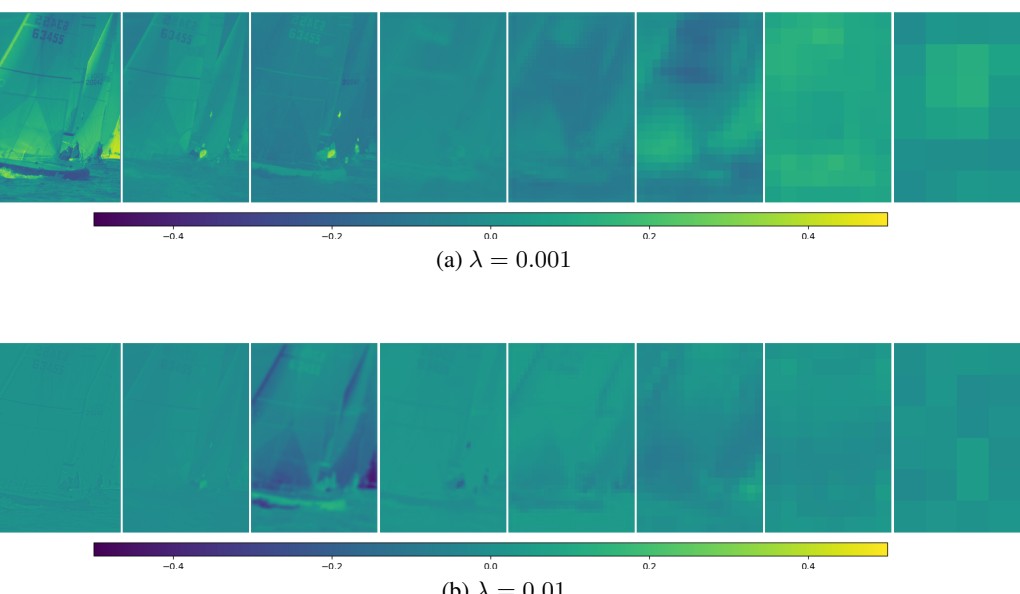

(a) $\lambda = 0.001$

(b) $\lambda = 0.01$

Figure 8: Visualizations of latents. These figures demonstrate normalized value for each latent.

Another ablation study explored the position of $\text{MLP}^i$. We investigated placing the $\text{MLP}^i$ at the input layer, intermediate layer, and output layer of the synthesis network, respectively. Table 3 presents the result. We found placing $\text{MLP}^i$ at the end of synthesis network will obtain the best performance.

Furthermore, the width of the synthesis network is also a parameter that needs to be examined. The Table 4 displays the performance under different widths, while the Fig. 9 illustrates the decoding speed performance at different widths. Overall, the configuration with a width of 16 is the optimal choice, taking into account decoding speed, RD performance, and hardware compatibility.

Table 2: Ablation result of RLE. RLE on 0 achieves the best performance among four settings. BD-Rate is calculated relative to RLE on 0.

| Setting | BD-Rate($\downarrow$) |
|---|---|
| No RLE | 150.68% |
| RLE on 0 | **0%** |
| RLE on 0, 1 | 1.18% |
| RLE on -1, 0, 1 | 2.70% |

Table 3: Start means we place $\text{MLP}^i$ at the beginning of the synthesis network. Middle means we place an $\text{MLP}^i$ between two shared MLPs. BD-Rate is calculated relative to default configuration, in which $\text{MLP}^i$ are placed at the end of synthesis network.

| Location | BD-Rate ($\downarrow$) |
|---|---|
| Start | 21.46% |
| Middle | 2.43% |
| End (default for PIC) | **0%** |

Table 4: Ablation result of the synthesis network width. BD-Rate is calculated relative to width 16.

| Width | BD-Rate($\downarrow$) |
|---|---|
| 8 | 3.03% |
| 16 | **0%** |
| 32 | 5.84% |

Fig. 10 uses BD-rate as the vertical axis to compare the encoding/decoding speed and decoding complexity of different methods. While our method does not achieve optimal performance across all dimensions, it demonstrates balanced capabilities with no significant weaknesses in the three metrics. In contrast, autoencoder methods offer fast encoding capabilities but suffer from high complexity, whereas Cool-chic maintains low complexity at the cost of significantly slower encoding and decoding speeds.

### A.3.4 ABLATION OF MODULATION

In addition to exploring the effect of the modulation mechanism on the model architecture proposed in this paper, we also validated the effectiveness of this mechanism on model architectures similar to N-O Cool-Chic. Fig. 11 illustrates the results of the ablation results. Since the last two layers of the N-O Cool-Chic's synthesis network are convolutional layers, we applied modulation to the features between the first two linear layers. The results show the effectiveness of the modulation mechanism in this architecture as well. Moreover, N-O Cool-Chic* with modulation has surpassed the method proposed in this paper in terms of RD performance, highlighting the potential of the end-to-end INR paradigm for further advancements in its network architecture and training recipts.

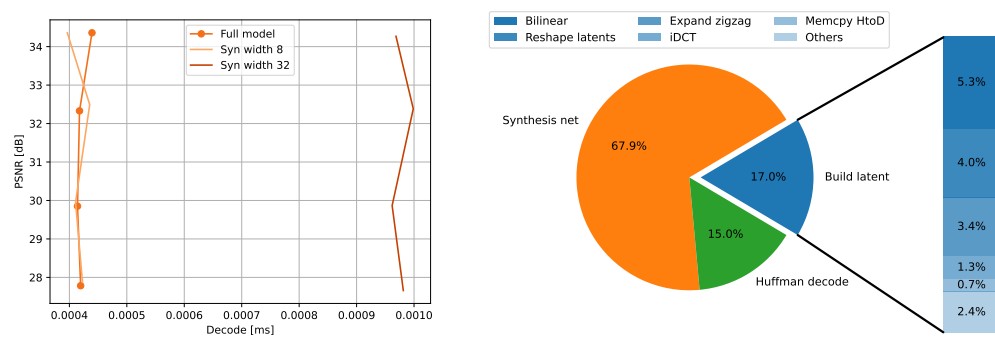

(a) Ablation results of decoding speed

(b) Detailed breakdown of decoding time

Figure 9: Fig. 9a is the ablation results of decoding speed for different width of synthesis network. Since the minimum matrix size supported by Tensor Cores for the TF32 data type is currently 16, a $16 \times 16$ matrix padded with zeros is used when the width is 8. Under this configuration, there is no significant difference in decoding speed between a width of 8 and a width of 16. Fig. 9b is detailed breakdown of decoding time including host-to-device I/O time.

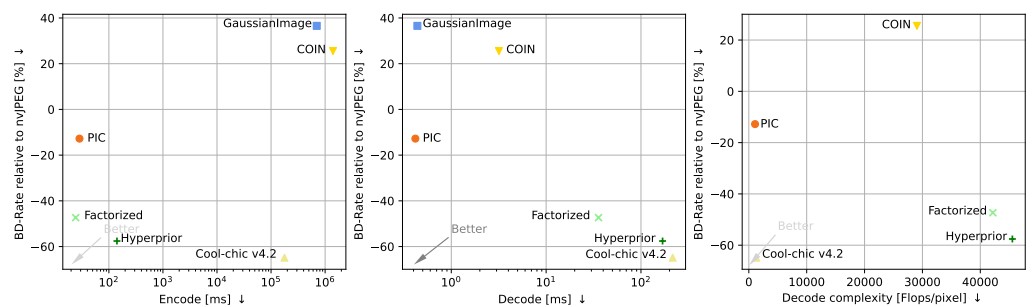

Figure 10: Results of comprehensive comparesion for both RD performance and practical performance on Kodak dataset.

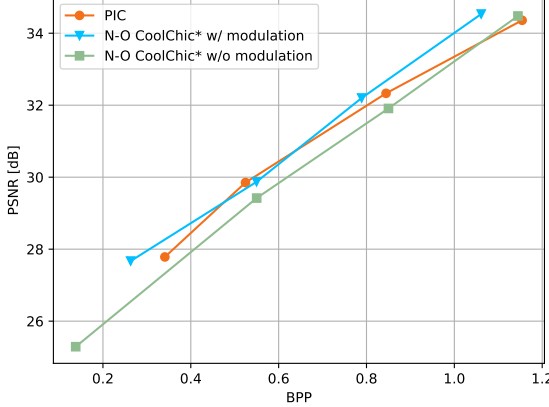

Figure 11: Ablation results of modulation mechanism on N-O Cool-Chic-like (mark by *) architecture. It should also be noted that N-O Cool-Chic is not open-source, so the results presented here are only a preliminary reproduction without hyperparameters and training recipts tuning and may differ from those reported in the original paper.