# OpenReview forum: "PIC: Revisiting INR for Image Coding with Fast Encoding and Sub-Millisecond Decoding"
_ICLR.cc/2026/Conference — Submitted to ICLR 2026_

### Official Review · Reviewer_6zum · 2025-10-24

**Soundness:** 3
**Presentation:** 2
**Contribution:** 3
**Rating:** 6
**Confidence:** 5

**Summary:**

This paper addresses the trade-off of Implicit Neural Representation (INR)-based image compression—low decoding complexity but slow encoding—and limitations of end-to-end models/JPEG. It proposes PIC (Practical INR Image Codec), an end-to-end architecture that generates all INR network parameters via a single forward pass, achieving 20 FPS encoding.

**Strengths:**

1. Transforms INR encoding from time-consuming per-image training to real-time single forward pass, boosting encoding speed by 3+ orders of magnitude vs. COIN/GaussianImage.

2. Aligns synthesis network (gs) with Tensor Core’s structure (16-channel layers) and uses Slang for training-inference consistency, enabling ultra-fast decoding.

3. ZpR module (simplified RLE on zeros) cuts bitrate without extra inference overhead.

**Weaknesses:**

1. While outperforming representation models, it still lags behind some end-to-end models.

2. Ablation studies show that shifting $MLP^i$ (instance-dependent part) from the end of $g_s$ to the start or middle drastically degrades performance (BD-Rate increases by 21.46% and 2.43%, respectively). This high sensitivity reduces architectural flexibility for further optimizations.

3. The MLP parameters in the bitstream are stored in FP32 format. Without lower-precision quantization (e.g., FP16), this increases bitstream size unnecessarily, compromising overall coding efficiency compared to models with parameter quantization.

4. More details in Figure 2 maybe better since I can not identify the calculatio preocess from the middle to the right.

**Questions:**

See Weaknesses and I would prefer to improve my rating for the rebuttal.

---

> ### Author Response · Authors · 2025-11-24
>
> We sincerely express our gratitude for dedicating your valuable time to providing insightful suggestions and positive feedback on our work. Your appreciation and acknowledgment of our manuscripts have greatly inspired us. Our detailed responses to all your questions are provided below.
>
> **W1**: While outperforming representation models, it still lags behind some end-to-end models.
>
> **R1**: Thanks for your suggestion. While our method does not achieve optimal RD performance, it should be noted that PIC demonstrates balanced capabilities with no significant weaknesses in comprehensive performance for practical applications. In contrast, autoencoder methods offer fast encoding capabilities but suffer from high complexity, whereas Cool-chic maintains low complexity at the cost of significantly slower encoding and decoding speeds.
>
> **W2**: Ablation studies show that shifting  (instance-dependent part) from the end of  to the start or middle drastically degrades performance (BD-Rate increases by 21.46% and 2.43%, respectively). This high sensitivity reduces architectural flexibility for further optimizations.
>
> **R2**: Thanks for your suggestion. This is indeed a question worth exploring. We will further investigate the reasons behind this phenomenon and seek better solutions in the future. Furthermore, as an additional supplementary experiment, we explored applying the proposed modulation mechanism to an N-O Cool-chic-like model [1]. The modified model achieved superior RD performance compared to PIC. The results are presented in the Fig. 11 in revised version. This demonstrates that the modulation mechanism proposed in this paper can be effectively applied to improve similar methods.
>
> **W3**: The MLP parameters in the bitstream are stored in FP32 format. Without lower-precision quantization (e.g., FP16), this increases bitstream size unnecessarily, compromising overall coding efficiency compared to models with parameter quantization.
>
> **R3**: Thanks for your suggestion. Using lower precision can reduce the bitrate of the network weight parameters. In the future, we will explore with even lower precision representations, such as bfloat16 and other formats, to further improve performance. Additionally, we will explore other network architectures to achieve better reconstruction quality under limited parameter constraints.
>
>
>
> **W4**: More details in Figure 2 maybe better since I can not identify the calculatio preocess from the middle to the right.
>
> **R4**: Thanks for your suggestion. We have added guidance information in Fig. 1 to enable readers to grasp the overall architecture of the method more easily.
>
> Thank you again for your valuable feedback! If you have any further questions or suggestions, please don't hesitate to tell us.
>
> **Reference**:
>
> [1] Blard, Théophile, et al. "Overfitted image coding at reduced complexity."

---

### Official Review · Reviewer_8sLj · 2025-10-28

**Soundness:** 3
**Presentation:** 2
**Contribution:** 3
**Rating:** 4
**Confidence:** 5

**Summary:**

In this work, the authors propose PIC, a new INR-based learned image compression framework that enables both fast encoding and decoding. Unlike most existing approaches that require overfitting the representation during encoding, PIC uses an INR whose layer weights are trained, while the grid parameters and activation modulation are generated by the encoder. The whole framework is trained end-to-end and requires only a forward pass during encoding/decoding. The encoding/decoding speed is very fast, and the RD performance is better than prior work including COIN and GaussianImage.

**Strengths:**

-	The framework is simple but effective for image coding. It is similar to a COOL-CHIC-like architecture, but uses the encoder to generate the modulation and grid parameters. It also benefits from training on a large dataset.
-	The decoder implementation is also an important contribution. There are very few works that implement learned compression models with optimized components such as custom GPU kernels, which is important for real-world applications.

**Weaknesses:**

- The compression performance is clearly not as good as common baseline methods, for example, the standard end-to-end models like the hyper-prior model, or more recent INR models like COOL-CHIC. It is not clear if the proposed method acheive a better trade-off between complexity and RD performance
- The comparison is limited.
    - The baselines are relatively weak in RD performance.
    - The authors should consider comparing to N-O COOL-CHIC [1], which also has very low complexity and is not overfitted.
- The framework does not seem strictly end-to-end, due to the use of the ZrP module. Also, the exact method for estimating the entropy is not given in the paper.
- There are some missing details in the paper
    - How is decoding done? Does it take the coordinates as input? Does it follow COOL-CHIC-style decoding? The authors should also consider providing a reference in this part, as this is not the main contribution of the paper.
    - How is $ p_{\theta_{k}}$ estimated?
    - What resolution is used in Table 1?
- While the main advantage of the proposed method is the encoding/decoding speed, the ablation study is focused on rate–distortion performance. Providing an ablation study on the encoding/decoding speed would be helpful.

[1] Blard, Théophile, et al. "Overfitted image coding at reduced complexity."

**Questions:**

- Why is the convolutional layer $g_{m}$ output size is $N \times C$? Are there some layers missing?
- How scalable is the framework? Given the optimized kernel for the tensor core, would increasing the network layer size cause a dramatic increase in decoding time?
- Line 292: How much does using full FP32 for training impact the performance? In my experience, many image codecs trained in FP32 work fine in FP16, and the error from FP32 to TF32 should be even smaller.

---

> ### Author Response · Authors · 2025-11-24
>
> We sincerely express our gratitude for dedicating your valuable time to providing insightful suggestions that can enhance our paper. We have carefully read all the comments and provide detailed point-by-point responses as follows. Hopefully, we can adequately address your concerns.
>
>
> **W1**: The compression performance is clearly not as good as common baseline methods, for example, the standard end-to-end models like the hyper-prior model, or more recent INR models like COOL-CHIC. It is not clear if the proposed method acheive a better trade-off between complexity and RD performance
>
> **R1**: Thanks for your suggestion. Regarding the selection of baseline methods, we primarily chose approaches with comparable decoding speed and complexity as our main points of comparison. The revised version has expanded its comparisons to include Cool-chic v4.2 (current SOTA for INR image codec) and the hyperprior model. We will add results on CLIC as soon as possible.  It should be noted that while our method does not achieve optimal RD performance, it demonstrates balanced capabilities with no significant weaknesses in comprehensive performance for practical applications. In contrast, autoencoder methods offer fast encoding capabilities but suffer from high complexity, whereas Cool-chic maintains low complexity at the cost of significantly slower encoding and decoding speeds.
>
>
> **W2**:The comparison is limited.
>
> **R2**: Thanks for your suggestion. We have added more baseline in revised version. Since N-O Cool-chic itself is not open-sourced, we were unable to fully reproduce its results within rebuttal period. Given the architectural similarities with Cool-Chic, it is reasonable to assume that its metrics (such as decoding speed and complexity) are comparable to those of Cool-chic. The results reported in this paper should therefore be considered as a reference. In terms of RD performance, N-O Cool-chic is expected to surpass PIC but underperform the Hyperprior according original paper. Its encoding speed is likely similar to PIC, while its decoding speed is estimated to be approximately $400\times$ slower than PIC. Furthermore, as an additional supplementary experiment, we explored applying the proposed modulation mechanism to an N-O Cool-chic-like model. The modified model achieved superior RD performance compared to PIC. The results are presented in the Fig. 11 in revised version. This demonstrates that the modulation mechanism proposed in this paper can be effectively applied to improve similar methods.
>
> **W3**: The framework does not seem strictly end-to-end, due to the use of the ZrP module. Also, the exact method for estimating the entropy is not given in the paper.
>
> **R3**:  Thanks for your suggestion. This non-fully-differentiable characteristic represents one of the limitations in the current architecture. We have also analyzed the discrepancy between the estimated and actual bit rates in Fig. 5b. In the future, we will further explore methods to reduce this gap and investigate more architectural variants, such as N-O Cool-chic.
>
> For entropy estimation, we employed the same entropy model as the Factorized [1] model, which is implmented in CompressAI [2]. This entropy model represented each of the marginals $p_{\theta_k}$ as a piecewise linear function. $\theta_k$ are updated simultaneously during training.
>
> **W4.1** How is decoding done? Does it take the coordinates as input? Does it follow COOL-CHIC-style decoding? The authors should also consider providing a reference in this part, as this is not the main contribution of the paper.
>
> **R4.1**: Thanks for your suggestion. Our method adopts an approach similar to the COOL-CHIC architecture in the reconstruction stage. However, in the latent decoding stage, instead of using the ARM mode from, we directly employs Huffman decoding to obtain all latents. We have add more description is revised version.
>
>
> **W4.2**: How is $p_{\theta_k}$ estimated?
>
> **R4.2**: Please refer to R3
>
> **W4.3**: What resolution is used in Table 1?
>
> **R4.3**: All metrics in Table 1 were evaluated on the Kodak dataset at a resolution of 512x768. Complexity is avaraged by pixels.
>
> **W5**: While the main advantage of the proposed method is the encoding/decoding speed, the ablation study is focused on rate–distortion performance. Providing an ablation study on the encoding/decoding speed would be helpful.
>
> **R5**: Thanks for your suggestion. We have added the results in Fig 9a in revised version.

---

> ### Author Response · Authors · 2025-11-24
>
> **Q1**: Why is the convolutional layer $g_m$ output size is $N\times C$? Are there some layers missing?
>
> **A1**:  Thanks for your question.$\gamma$ and $\beta$ are mean and standard deviation of each channel of $g_m$ output respectively. We have added the illustration in revised version.
>
> **Q2**: How scalable is the framework? Given the optimized kernel for the tensor core, would increasing the network layer size cause a dramatic increase in decoding time?
>
> **A2**:  Thanks for your question. We have added more ablation results for synthesis net width in Fig. 9a and Table 4. Under the current architecture, increasing or decreasing the width of the synthesis net does not yield a significant performance improvement. Increasing the network width will also lead to a significant increase in decoding time. In terms of the overall design, current experimental results indicate that there is still room for architectural refinement; however, simply increasing the model scale does not guarantee performance gains.
>
>
> **Q3**: Line 292: How much does using full FP32 for training impact the performance? In my experience, many image codecs trained in FP32 work fine in FP16, and the error from FP32 to TF32 should be even smaller.
>
> **A3**: Thanks for your question. The  performance gap between Float32 and TF32 during training and inference is identified in our experiments. We hypothesize that a neural network's robustness to parameter precision is dependent on model size or overparameterization. Given that our synthesis is a very compact model, simply altering the data type leads to significant discrepancies in results. Therefore, it is crucial to maintain consistency in data types across both training and testing phases.
>
> We hope these replies help to clarify your concerns. We truly appreciate your constructive suggestions for improving our work and would be happy to discuss this with you further.
>
> **Reference**:
>
> [1] Johannes Ballé, et al. "End-to-end Optimized Image Compression."
>
> [2] Jean Bégaint, et al. "CompressAI: a PyTorch library and evaluation platform for end-to-end compression research."

---

### Official Review · Reviewer_RCpp · 2025-10-29

**Soundness:** 2
**Presentation:** 3
**Contribution:** 2
**Rating:** 4
**Confidence:** 5

**Summary:**

The proposed PIC is an E2E feed-forward INR-based image codec, where a neural encoder produces all parameters of a tiny per-image decoder in one forward pass, based on channel-wise affine modulations, thereby circumventing the long training typical of INR codecs. The codec also features a lightweight entropy path (channel-wise), a simplified pRLE (where run-length is on zeros only), and a highly optimized GPU decoder implementation. It reports ~20FPS encode and ~2000FPS decode on a single 4090-class GPU, decoding faster than nvJPEG while being comparable in RD performance.

**Strengths:**

- The paper closes the INR practicality gap with its pure feed-forward encoding (no per-image training at all) and ultra-fast decoding. Only GaussianImage and the proposed PIC, to the best of my knowledge, could decode images at the magnitude of ~2000 FPS, yet PIC is stronger than GaussianImage.
- The compression performance is comparable against JPEG at higher bpp ranges on Kodak, compared to COIN and GaussianImage.
- The experimental results are complete, self-contained, with fruitful and honest engineering details: careful timing protocol, CUDA kernels for ZpR/decoder, Slang-Torch implementation notes, etc.

**Weaknesses:**

- **Unfair profiling.** JPEG is normally I/O-bound and optimized for batch streaming, but the paper measures it in-process without I/O, a certainly fair but probably atypical setup. Since PIC’s decoding pipeline already runs fully in-memory, removing I/O overhead disproportionately benefits PIC, making the comparison slightly optimistic. I would recommend the authors provide a more detailed breakdown of the per-component runtimes.
- **Entropy estimation vs actual drift.** Authors show noticeable gaps between estimated and realized bitrate (Pearson around 0.95–0.99 depending on dataset and $\lambda$); they note this is common but it indeed complicates precise rate control.
- **Compression efficiency.** I think the selected baselines are relatively weak, and that the compression performance is not standing out either. In my person opinion, I am not entirely sure if 2000FPS is that necessary (maybe the difference while be more noticeable on higher resolution images?), but a performance only comparable to JPEG is not good enough.
- **Scalability.** The method appears tightly coupled to a fixed channel width (C = 16): every layer in the analysis and synthesis paths uses this width, and the custom CUDA decoder kernel seems specifically tuned to 16×16 Tensor Core tiles. This raises concerns about scalability and generality: it’s unclear whether PIC maintains speed or compression efficiency when the channel dimension changes, or if the implementation would require re-engineering to handle higher-capacity models. Could the authors clarify whether the architecture and fused kernels support other channel sizes (e.g., 8, 32, 64) and provide throughput/RD results for such variants to show the design scales beyond a single configuration?

Overall, I think the paper is of good quality with clear motivation and strong engineering, but its technical novelty and compression performance remain limited. I would consider raising my score if the authors provide evidence of scalability (beyond C=16), stronger RD comparisons against modern learned codecs (or counter-arguments to why this is not compulsory), and a more complete runtime breakdown that includes I/O and decoding components to substantiate the claimed efficiency advantage.

**Questions:**

- I think the paper would benefit from a rate-distortion-complexity frontier comparison (like Figure 1 of MobileNVC [1]) and that PIC should clearly demonstrate a consistently better performance (manifested as a better Pareto frontier) than existing baselines.

[1] MobileNVC: Real-time 1080p Neural Video Compression on a Mobile Device, WACV'24.

---

> ### Author Response · Authors · 2025-11-24
>
> We sincerely express our gratitude for dedicating your valuable time to providing insightful suggestions that can enhance our paper. Your appreciation regarding our quality has greatly encouraged us. Our detailed responses to all of your concerns are presented below.
>
>
> **W1**: **Unfair profiling.** JPEG is normally I/O-bound and optimized for batch streaming, but the paper measures it in-process without I/O, a certainly fair but probably atypical setup. Since PIC’s decoding pipeline already runs fully in-memory, removing I/O overhead disproportionately benefits PIC, making the comparison slightly optimistic. I would recommend the authors provide a more detailed breakdown of the per-component runtimes.
>
> **R1**: Thanks for your suggestion. In our evaluation, we have made every effort to eliminate the impact of various factors on fairness. As you mentioned, JPEG is I/O-bound, so we minimized this influence by adopting in-memory decoding for both NVJPEG and the proposed PIC. Furthermore, we ensured that the input bitstream during the decoding phase originates from data residing on the CPU (py::bytes). We believe this approach can minimize the differences in I/O measurement between the two methods. If there are any oversights in our consideration, we welcome your feedback on the potential issues.
>
> Figure 9b in the revised version provides a detailed breakdown of the time consumption for each module within the PIC decoder. Since NVJPEG is a closed-source implementation, conducting a more in-depth analysis presents certain challenges. Based on publicly available information, we speculate empirically that the  "Hybrid decoding using both the CPU and the GPU" slowed down the decoding speed of NVJPEG. Because our decoder, on the other hand, is a pure GPU implementation. This may lead to differences in the amount of data transferred between the CPU and GPU and the speed of Huffman decoding. Regarding batch decoding, although our current tests employ a serial decoding design, adapting it to batch decoding would not pose theoretical challenges. Furthermore, due to the similarities in decoding architecture, hardware acceleration circuits originally designed for JPEG could, in theory, also be leveraged to further improve the decoding speed of our method.
>
> **W2**: **Entropy estimation vs actual drift.** Authors show noticeable gaps between estimated and realized bitrate (Pearson around 0.95–0.99 depending on dataset and ); they note this is common but it indeed complicates precise rate control.
>
> **R2**: Thank you for your question. The current rate constraint approach has largely achieved its intended purpose, though the drift from the true values does have some impact on the final outcome. Based on our research, there are currently no precise entropy estimation methods specifically designed for codecs like JPEG. In the future, we will strive to make efforts in this direction to reduce the discrepancy between estimated and actual values.
> Additionally, we will explore the alternative entropy model architectures in future.
>
> **W3**: **Compression efficiency.** I think the selected baselines are relatively weak, and that the compression performance is not standing out either. In my person opinion, I am not entirely sure if 2000FPS is that necessary (maybe the difference while be more noticeable on higher resolution images?), but a performance only comparable to JPEG is not good enough.
>
> **R3**: Thank you for your suggestion. Regarding the selection of baseline methods, we primarily chose approaches with comparable decoding speed and complexity as our main points of comparison. The revised version has expanded its comparisons to include Cool-chic v4.2 (current SOTA for INR image codec) and the hyperprior model. We will add results on CLIC as soon as possible.
>
> In terms of decoding speed, the 2000 FPS achieved in this paper may indeed be unnecessary for most practical scenarios. However, as the first work on end-to-end INR image coding, this result establishes a milestone and defining characteristic for this paradigm. For future work, this design could be relaxed to improve RD performance. For instance, employ more complex synthesis networks to better adapt to the multi-dimensional trade-offs between rate, distortion, complexity, and speed required in real-world applications.

---

> ### Author Response · Authors · 2025-11-24
>
> **W4**: **Scalability.** The method appears tightly coupled to a fixed channel width (C = 16): every layer in the analysis and synthesis paths uses this width, and the custom CUDA decoder kernel seems specifically tuned to 16×16 Tensor Core tiles. This raises concerns about scalability and generality: it’s unclear whether PIC maintains speed or compression efficiency when the channel dimension changes, or if the implementation would require re-engineering to handle higher-capacity models. Could the authors clarify whether the architecture and fused kernels support other channel sizes (e.g., 8, 32, 64) and provide throughput/RD results for such variants to show the design scales beyond a single configuration?
>
> **R4**: Thank you for your suggestion. We have added more ablation results in revised version. In Appendix 3.3 and Table 4, we demonstrate different setting of width of synthesis net. The results shows increasing or reducing the size of the network does not lead to performance improvement. Fig 9a shows the decoding speed of these configurations.
>
> Regarding the architecture of the encoding/analysis network, we consider it a non-trivial problem that merits further in-depth investigation, and we will focus on this aspect in the future. As an initial result, combining our proposed modulation mechanism with N-O Cool-chic-like model has achieved superior RD performance compared to PIC. The experimental results are presented in the Fig. 11 in revised version.
>
> These two sets of experimental results also indicate that such paradigm still requires the exploration of superior network architectures in the future, as simply increasing the model size does not necessarily lead to improved performance.
>
> **Q1**: I think the paper would benefit from a rate-distortion-complexity frontier comparison (like Figure 1 of MobileNVC ) and that PIC should clearly demonstrate a consistently better performance (manifested as a better Pareto frontier) than existing baselines.
>
> **A1**: Thanks for your question. We have added the result to Fig 10 in revised version. It should be noted that while our method does not achieve optimal performance across all dimensions, it demonstrates balanced capabilities with no significant weaknesses in the three metrics. In contrast, autoencoder methods offer fast encoding capabilities but suffer from high complexity, whereas Cool-chic maintains low complexity at the cost of significantly slower encoding and decoding speeds.
>
> Thank you again for your insightful suggestion. If you have any further question, feel free contact us.

---

### Official Review · Reviewer_a2mu · 2025-11-05

**Soundness:** 2
**Presentation:** 2
**Contribution:** 3
**Rating:** 4
**Confidence:** 4

**Summary:**

This paper presents an image compression framework that uses an implicit neural‐representation style approach but replaces instance‐specific training with a one‐pass encoding network, to achieve fast encoding and ultra-fast decoding while maintaining competitive rate‐distortion performance. For the one-pass encoding, the authors relied on the end to end compression framework, extracted the hierarchical latents and entropic coded in the DCT domain, and trained the modulation network and light weight synthesis network parameters.   The authors demonstrate encoding at ~20 fps and decoding at ~2000 fps, and show competitive RD vs JPEG and compared methods on Kodak/CLIC.

**Strengths:**

1) The major strength of this paper is the encoding and decoding speed, by taking the advantage of both end to end compression method and implicit neural representation.
2) The method is carefully implemented, integrating DCT transforms, entropy modeling, and low-level hardware optimizations (Tensor Cores, Slang-Torch).

**Weaknesses:**

Following are the weakness of the paper

1) The authors mostly focus on the encoding and decoding complexity of the neural codec, and they were able to propose method which is optimal in both, and the RD performance only matches or outperform in margin of JPEG. Most of the recent traditional codecs has higher RD performance with slightly higher complexity than JPEG. The RD performance of the proposed method compared to the learned codec is very low. The trade-off of complexity, rate and distortion needs to reasonable for the practical usage, the paper only outperform in one axis (complexity), and on other two axis (rate and distortion) needs to be improved.

2) The contribution of the paper is more of engineered solution to adapt the end to end codec for the INR codec.

3) Without the close performance to the Factorized prior model, I believe the proposed method does not offer any additional advantage over the JPEG

4) The authors description "encoding in one step" is somewhat misleading, the latents are generated in one-step, but the entropy model and modulation network, synthesis network requires atleast iterations to optimize.

**Questions:**

1) To optimize the modulation network, entropy model, and the synthesis network how many training iterations are performed? If low number of iterations are used, whether by increasing training iterations the performance could be improved.

2) how the common part of the synthesis network is trained, the details are missing?. There are some works [1]  in literature of the end to end compression, whether the part of the synthesis network is fine-tuned for a given image/signal, how this is different/similar in the paper?

[1] Improving The Reconstruction Quality by Overfitted Decoder Bias in Neural Image Compression, https://ieeexplore.ieee.org/abstract/document/10018052
[2] Overfitting for Fun and Profit: Instance-Adaptive Data Compression, https://arxiv.org/abs/2101.08687

---

> ### Author Response · Authors · 2025-11-24
>
> We sincerely thank you for your precious time and effort in providing a wealth of suggestions to enhance the quality of our paper. We have carefully read all the comments and provide detailed point-by-point responses as follows. Hopefully, we can adequately address your concerns.

---

> ### Author Response · Authors · 2025-11-24
>
> **W1**: The authors mostly focus on the encoding and decoding complexity of the neural codec, and they were able to propose method which is optimal in both, and the RD performance only matches or outperform in margin of JPEG. Most of the recent traditional codecs has higher RD performance with slightly higher complexity than JPEG. The RD performance of the proposed method compared to the learned codec is very low. The trade-off of complexity, rate and distortion needs to reasonable for the practical usage, the paper only outperform in one axis (complexity), and on other two axis (rate and distortion) needs to be improved.
>
> **R1**: Thank you for your suggestion. As you pointed out, many current neural encoders demonstrate outstanding performance in RD (Rate-Distortion) performance. However, the high complexity of such methods prevents them from being applied in real-world scenarios. In contrast, the complexity advantages and fast encoding-decoding characteristics of our method make it practical, which is the main reason we proposed this approach. We have also discussed this in our paper. Although our method does not match the RD performance of the most advanced neural encoders, considering that our paper is the first to explore this approach and the network structure design is relatively preliminary, I believe there is significant room for future improvement.
>
> **W2**: The contribution of the paper is more of engineered solution to adapt the end to end codec for the INR codec.
>
> **R2**: Thank you for your suggestion. Generating INR weights end-to-end is a non-trivial challenge. In our ablation studies, we compared variants based on our proposed generation architecture, and we also believe there are still some architectural possibilities we have not yet explored.  Furthermore, as an additional supplementary experiment, we explored applying the proposed modulation mechanism to an N-O Cool-chic-like model [1]. The modified model achieved superior RD performance compared to PIC. The results are presented in the Fig. 11 in revised version. This demonstrates that the modulation mechanism proposed in this paper can be effectively applied to improve similar methods. We are confident that our exploration of end-to-end INR generation network structures can bring new insights to the community.
>
> **W3**: Without the close performance to the Factorized prior model, I believe the proposed method does not offer any additional advantage over the JPEG
>
> **R3**: Thank you for your question. Currently, the RD performance is a limitation of our method. However, this work represents a preliminary exploration that employs a very simple network architecture. We plan to investigate other more efficient structures in the future. Considering the comprehensive dimensional performance characteristics of our approach, we believe it serves as a solid starting point.

---

> ### Author Response · Authors · 2025-11-24
>
> **W4 & Q1**: The authors description "encoding in one step" is somewhat misleading, the latents are generated in one-step, but the entropy model and modulation network, synthesis network requires atleast iterations to optimize. / To optimize the modulation network, entropy model, and the synthesis network how many training iterations are performed? If low number of iterations are used, whether by increasing training iterations the performance could be improved.
>
> **R4**: Thank you for your question. There might be some misunderstanding here. Our method indeed obtains all the information of the final INR, including latents and the synthesis net, through a single forward pass. As illustrated in Fig. 1, the weight, or more specific the modulation info, of $g_s$ is generated by $g_m$. Since the entropy model itself is parameter-free, there is no need to obtain corresponding parameters. Therefore, there is no further fine-tuning stage during the encoding process. From another perspective, if such operations were to be introduced, our method would not be able to achieve fast encoding.
>
> **Q2**: 1. how the common part of the synthesis network is trained, the details are missing?. There are some works in literature of the end to end compression, whether the part of the synthesis network is fine-tuned for a given image/signal, how this is different/similar in the paper?
>
> **A2**: Thank you for your question. The weights of the entire network, including the common part in the synthesis net, are trained in an end-to-end manner and do not involve any per-image fine-tuning process. The two articles you mentioned require per-image fine-tuning during the encoding phase to achieve high efficiency, whereas our method does not. In other words, although our approach ultimately generates an "overfitted" INR, its architecture and training pipeline are designed for generalizable codec. The only distinction lies in the output modality, which is not conventional images or videos but rather an INR.
>
> We hope these responses can address your concerns. Once again, we deeply appreciate your valuable suggestions for improving our work and would be delighted to further discuss with you.
>
> **Reference**:
>
> [1] Blard, Théophile, et al. "Overfitted image coding at reduced complexity."

---

### Meta-Review · Area_Chair_edf6 · 2026-01-10

**Summary:**

The paper receives mostly negative reviews (4, 4, 4, 6). Overall, the rebuttal responses from the authors are not convincing enough and the paper needs more work, particularly in improving its RD performance. Currently, its RD performance is not attractive at all as compared to other competing methods although the authors argue that their goal is to strike a better balance between RD performance and complexity and that their idea can benefit more capable codecs. But, more evidence and work is required.

Reviewer a2mu (4: marginally below the acceptance threshold. But would not mind if paper is accepted; 4: You are confident in your assessment.)

o	The authors mostly focus on the encoding and decoding complexity of the neural codec, and they were able to propose method which is optimal in both, and the RD performance only matches or outperform in margin of JPEG. Most of the recent traditional codecs has higher RD performance with slightly higher complexity than JPEG. The RD performance of the proposed method compared to the learned codec is very low. The trade-off of complexity, rate and distortion needs to reasonable for the practical usage, the paper only outperform in one axis (complexity), and on other two axis (rate and distortion) needs to be improved.

[AC: The authors admit the inferior performance of their approach as compared to the baseline methods but also argue that they are the first to explore this approach and their network structure design is relatively preliminary. I believe this paper needs more work to be considered for publication.]

o	The contribution of the paper is more of engineered solution to adapt the end to end codec for the INR codec.

[AC: Additional results are generated during the rebuttal period to show that the proposed modulation mechanism can benefit N-O Cool-chic-like models. Again, how it performs as compared to JPEG and VVC intra coding is very critical.]

o	Without the close performance to the Factorized prior model, I believe the proposed method does not offer any additional advantage over the JPEG

[AC: The authors admit that the current limitation of their work is relatively poor RD performance. However, this is critical for a compression paper to be considered for publication.]

o	The authors description "encoding in one step" is somewhat misleading, the latents are generated in one-step, but the entropy model and modulation network, synthesis network requires atleast iterations to optimize.
o	To optimize the modulation network, entropy model, and the synthesis network how many training iterations are performed? If low number of iterations are used, whether by increasing training iterations the performance could be improved.

[AC: These are clarification questions and are not so critical.]

Reviewer RCpp (4: marginally below the acceptance threshold. But would not mind if paper is accepted; 5: You are absolutely certain about your assessment.)

o	W1: Unfair profiling. JPEG is normally I/O-bound and optimized for batch streaming, but the paper measures it in-process without I/O, a certainly fair but probably atypical setup. Since PIC’s decoding pipeline already runs fully in-memory, removing I/O overhead disproportionately benefits PIC, making the comparison slightly optimistic. I would recommend the authors provide a more detailed breakdown of the per-component runtimes.

[AC: The authors argue that in-memory decoding is performed for both NVJPEG and the proposed PIC. Moreover, they indicate that the input bitstream during the decoding phase originates from data residing on the CPU (py::bytes). A breakdown analysis of the time consumption for each module within the PIC decoder was additionally provided during the rebuttal period in Fig. 9b. However, the authors also admit that “Based on publicly available information, we speculate empirically that the "Hybrid decoding using both the CPU and the GPU" slowed down the decoding speed of NVJPEG. Because our decoder, on the other hand, is a pure GPU implementation.” It then becomes a bit inconclusive.]

o	Entropy estimation vs actual drift. Authors show noticeable gaps between estimated and realized bitrate (Pearson around 0.95–0.99 depending on dataset and ); they note this is common but it indeed complicates precise rate control.

[AC: The authors argue that there is no precise rate estimation scheme for JPEG (which I kind of agree with), and will explore other entropy models. ]

o	Compression efficiency. I think the selected baselines are relatively weak, and that the compression performance is not standing out either. In my person opinion, I am not entirely sure if 2000FPS is that necessary (maybe the difference while be more noticeable on higher resolution images?), but a performance only comparable to JPEG is not good enough.

[AC: Poor RD performance is the major concern from most reviewers, but at this point, I do not see an effective rebuttal response. This is considered by the authors to be their future work.]

o	Scalability. The method appears tightly coupled to a fixed channel width (C = 16): every layer in the analysis and synthesis paths uses this width, and the custom CUDA decoder kernel seems specifically tuned to 16×16 Tensor Core tiles. This raises concerns about scalability and generality: it’s unclear whether PIC maintains speed or compression efficiency when the channel dimension changes, or if the implementation would require re-engineering to handle higher-capacity models. Could the authors clarify whether the architecture and fused kernels support other channel sizes (e.g., 8, 32, 64) and provide throughput/RD results for such variants to show the design scales beyond a single configuration?

[AC: Additional results were provided during the rebuttal period in Appendix 3.3 and Table 4.  No performance improvement is observed by increasing or reducing the size of the network. Simply increasing the model size does not really help. Further investigation into superior network architectures is warranted.]

Reviewer 8sLj (4: marginally below the acceptance threshold. But would not mind if paper is accepted; 5: You are absolutely certain about your assessment.)
o	The compression performance is clearly not as good as common baseline methods, for example, the standard end-to-end models like the hyper-prior model, or more recent INR models like COOL-CHIC. It is not clear if the proposed method acheive a better trade-off between complexity and RD performance
o	The comparison is limited.
	The baselines are relatively weak in RD performance.
	The authors should consider comparing to N-O COOL-CHIC [1], which also has very low complexity and is not overfitted.

[AC: Additional comparison with Cool-chic v4.2 (current SOTA for INR image codec) and the hyperprior model was provided. Again, the proposed method does not achieve comparable RD performance to these more advanced models, although the authors argue that their approach strikes a good balance between coding performance and encoding/decoding complexity.]

o	 The framework does not seem strictly end-to-end, due to the use of the ZrP module. Also, the exact method for estimating the entropy is not given in the paper.

[AC: The authors admit that their framework is non-fully-differentiable, and they use the Factorized model for rate estimation.]

[AC: The other comments are clarification ones, and not so critical.]

Reviewer 6zum (6: marginally above the acceptance threshold. But would not mind if paper is rejected; 5: You are absolutely certain about your assessment.)

o	While outperforming representation models, it still lags behind some end-to-end models.

[AC: The same RD performance concern as the other reviewers’.]

o	Ablation studies show that shifting  (instance-dependent part) from the end of  to the start or middle drastically degrades performance (BD-Rate increases by 21.46% and 2.43%, respectively). This high sensitivity reduces architectural flexibility for further optimizations.

[AC: The authors argue that “This is indeed a question worth exploring. We will further investigate the reasons behind this phenomenon and seek better solutions in the future.” More work needs to be done.]

o	The MLP parameters in the bitstream are stored in FP32 format. Without lower-precision quantization (e.g., FP16), this increases bitstream size unnecessarily, compromising overall coding efficiency compared to models with parameter quantization.

[AC: The authors argue that “This is indeed a question worth exploring. We will further investigate the reasons behind this phenomenon and seek better solutions in the future.” Again, more work needs to be done.]

[AC: The other questions are clarification ones and not so critical.]

**Reviewer Concerns:**

See my AC comments in the summary section.

**Reviewer Scores:**

The paper receives mostly negative reviews (4, 4, 4, 6). Overall, the rebuttal responses from the authors are not convincing enough and the paper needs more work, particularly in improving its RD performance. Currently, its RD performance is not attractive at all as compared to other competing methods although the authors argue that their goal is to strike a better balance between RD performance and complexity and that their idea can benefit more capable codecs. But, more evidence and work is required. I would recommend "reject".

---

### Decision · Program_Chairs · 2026-01-26

Reject